# Perceived Benefits of an Adaptive Tai Chi Program Among Veterans with Ambulatory Limitations

**DOI:** 10.3390/ijerph22091326

**Published:** 2025-08-26

**Authors:** Zack Simoni, Darrell Walsh, Lori Waite, Beth Herring, Karen Wilson, Chang Phuong, Zibin Guo

**Affiliations:** 1Department of Sociology, Anthropology, and Geography, The University of Tennessee at Chattanooga, Chattanooga, TN 37403, USA; 2Independent Researcher, Chattanooga, TN 37403, USA; 3College of Engineering and Computer Science, The University of Tennessee at Chattanooga, Chattanooga, TN 37403, USA

**Keywords:** disability, adaptive Tai-Chi, veterans, ambulatory limitations, seniors, physical activity, social connection

## Abstract

Background: The growing population of aging veterans in the United States often experiences disabilities that restrict physical activity and limit overall well-being and self-reported health. Accessible, practical, and inclusive interventions are crucial to improve their well-being. Objectives: This study aimed to explore the perceived benefits of an adaptive Tai Chi program among veterans with ambulatory limitations. Methods: The researchers conducted a qualitative thematic analysis to thoroughly investigate veterans’ experiences and perceptions regarding an adaptive Tai Chi intervention. Results: Four primary benefits of adaptive Tai Chi emerged from the perspectives of the veteran participants. These included psychological improvements such as heightened mindfulness, enhanced emotional regulation, and a greater sense of control over thoughts and emotions, proving especially valuable for those managing PTSD. Additionally, the program fostered strong social connections and was perceived as highly inclusive, accommodating diverse physical abilities. We also find that the accommodating and adaptive nature of the program empowered veterans to reframe their disability and expand their perception of their physical capabilities. Conclusions: These detailed qualitative findings suggest that adaptive Tai Chi may be a valuable therapeutic intervention for improving the overall well-being of aging veterans with ambulatory challenges while also addressing their psychological, social, and physiological needs.

## 1. Introduction

The United States is experiencing a significant demographic shift with its growing aging population. As of 2024, there were more than 58 million adults aged 65 and older living in the United States, and adults aged over 65 account for nearly 18% of the nation’s total population [1]. As the aging population continues to grow in the United States, so does the prevalence of disability. Research indicates that 30% of adults over the age of 65 have one or more disabilities, and this number rises to nearly half (46%) for those aged 75 and older and over 75% for adults over the age of 85 [2]. Disabilities include a wide array of conditions involving mobility issues, cognitive impairments, chronic pain, and difficulties with independent living [3,4,5].

Veterans represent a particularly vulnerable subgroup within the aging population in the United States. In 2023, in the United States, almost 2.3 million veterans had a service-connected disability rating of 70 percent or higher [6]. Furthermore, veterans are almost twice as likely to have a disability compared to the general adult population (16%). Among veterans, a substantial proportion have service-connected disabilities, meaning their disability resulted from an injury or illness incurred during their military service; in August 2024, this accounted for 31% of all veterans [6]. The high prevalence of disability among aging veterans underscores the urgent need for accessible and effective interventions to increase participation in physical activity and enhance their physical and mental well-being and overall life satisfaction.

Despite the need for physical activity, the aging population, and those with disabilities and ambulatory limitations, face substantial barriers to engaging in physical activity. These challenges span physiological, psychological, social, and environmental domains [7,8]. Physiologically, researchers have consistently found age-related declines in musculoskeletal function, as well as joint stiffness, and various chronic conditions like arthritis and osteoporosis make movement painful and difficult, often leading to a fear of aggravating existing health issues [9,10,11,12,13]. Psychologically, previous research has identified that the fear of injury or falling, a perceived lack of ability, low self-efficacy, and a generalized belief of being “too old to exercise” can lead to a lack of motivation and avoidance of physical activity [7,11,13,14,15,16]. Socially, barriers include a lack of encouragement from family and peers, age-related stereotypes about ability, fear of judgment from others, and limited access to age-appropriate community programs—all of which contribute to isolation and reduced engagement [7,16,17].

Recognizing the importance of physical activity, various interventions have been implemented to promote engagement among individuals with disabilities and ambulatory limitations. These approaches comprise a range of strategies, including adaptive sports and recreation programs, which provide modified physical activities but may be limited by accessibility, equipment costs, and motivation to engage in competitive sports and may be difficult for those with specific types of disability [18,19]. Home-based exercise programs are practical solutions which use specialized equipment or tailored exercise routines to increase activity within an individual’s home [20,21]. They offer convenience but might suffer from a lack of supervision as well as limited motivation, limited technological literacy, a risk of falls, and adherence over the long term [22]. Wearable devices, virtual reality platforms, and telerehabilitation are also increasingly being explored to improve physical activity for this population, but may be limited by issues with digital literacy [23]. While these various interventions have demonstrated efficacy in promoting physical activity and improving various health outcomes, persistent challenges remain in achieving consistent and long-term adherence. Thus, there is an ongoing need for additional intervention strategies to increase physical activity among this population.

Considering these barriers, interventions like Tai Chi offer unique advantages. Tai Chi is an ancient mind–body practice that may offer an additional form of physical activity while addressing limitations of the interventions mentioned above [24,25]. Unlike many adaptive sports programs, Tai Chi is a low-cost, low-impact exercise that can be performed with minimal space and no equipment, making it highly accessible for individuals with ambulatory limitations who may face transportation or financial barriers [26]. Tai Chi can also be easily adapted to address mobility limitations; hence, this makes it suitable for a wide range of functional abilities [26,27]. The mindful and meditative aspects of Tai Chi may foster intrinsic motivation and improve adherence by enhancing body awareness [26]. Lastly, researchers have demonstrated Tai Chi’s efficacy in attenuating age-related decline, improving balance, and reducing falls among the aging population [25,28,29,30,31,32], and some research has shown enhanced cognitive functions in older adults, including those with dementia [33,34].

This study highlights the Wheelchair/Adaptive Tai Chi Chuan (W/A TCC) program, developed by Dr. Zibin Guo. The program provides adapted physical activity within an inclusive environment, emphasizing relaxation and control as mechanisms for managing stress and pain through adapted physical activity and social interaction for those with ambulatory limitations [35]. To implement this program, the Veterans Affairs (VA) Adaptive Sports grant supported dozens of instructional workshops at numerous VA medical centers, training over 800 healthcare providers and offering classes to thousands of veterans.

The W/A TCC program had been taught in person for most of its existence. However, the COVID-19 pandemic made in-person instruction impossible. As such, the classes were taught virtually for the safety of participants. Research indicates that online Tai Chi is feasible and effective, demonstrating improvements in physical function, balance, and mental well-being in various populations, including older adults [36]. The virtual modality inherently overcomes geographical and transportation barriers, while offering convenience and potentially improving long-term adherence for individuals who might otherwise be unable to participate in in-person programs.

There is currently a limited understanding of how the participants themselves experience and perceive the benefits of the (W/A TCC) program and Tai Chi more generally. Exploring these individual perspectives through qualitative research is essential because perceptions can directly influence veterans’ motivation to participate in the program, their adherence, and their overall experience of the program, ultimately potentially increasing their quality of life [26,27,35]. Therefore, this paper aims to explore the perceived benefits of adaptive Tai Chi from the perspectives of veterans with ambulatory concerns and limitations. We posit the following research questions: (a) What are the perceived psychological benefits of adaptive Tai Chi among aging veterans with ambulatory limitations? (b) What are the perceived physiological benefits of adaptive Tai Chi among aging veterans with ambulatory limitations? (c) What are the perceived social benefits of adaptive Tai Chi among aging veterans with ambulatory limitations?

## 2. Materials and Methods

We chose to use inductive thematic analysis as our methodology. Thematic analysis is a systematic research method used to identify and provide insight into patterns of meaning-making in the description of phenomena [37]. Beyond simply identifying patterns in behaviors, this approach enables the researcher to uncover ways of knowing and experiencing behaviors and their motivations [38]. As with many qualitative approaches, the goal is to understand the participants’ lived experience from their perspective in a highly detailed manner. Further, qualitative thematic analysis allows the flexibility to explore emerging themes as well as the exploration of novel research questions, especially when existing literature is insufficient to describe the specific research question(s). Thus, the approach fulfilled the goal of fully understanding participants’ experiences and the perceived benefits of the adaptive Tai Chi program.

### 2.1. Data Collection

Considering the goals of this study and its qualitative methodology, we decided to utilize purposive sampling as a sampling technique. Purposive sampling is a non-probability sampling technique in which researchers intentionally select participants based on specific characteristics, knowledge, experiences, or other criteria that are directly relevant to their research question and objectives [39]. In other words, the goal is to acquire “information rich” participants who can provide deep insights into the social phenomena being studied.

We selected participants based on the following inclusion criteria. First, they were veterans who had served in the United States military. Second, they reported experiencing ambulatory limitations or a disability throughout their daily lives as noted by instructors. Third, they participated in the online adaptive W/A Tai Chi Chuan (TCC) training class during 2023. All participants were referred to this adaptive Tai Chi program by their healthcare providers at their respective VA medical centers. According to communication with their healthcare providers, all participants had a disability, including emotional distress.

We developed the online classes—offered virtually through Zoom—as an extension of previous face-to-face sessions, particularly in response to the COVID-19 pandemic-related restrictions. The virtual modality of the program provided various benefits, including eliminating geographical and transportation issues, while simultaneously providing convenience and potentially enhancing long-term participation for individuals who might otherwise face barriers to in-person programs. Studies show that online Tai Chi is both feasible and effective, leading to improvements in physical function, balance, and mental well-being across diverse populations, including older adults [36].

Healthcare providers at various VA medical centers nationwide frequently recommended these virtual classes to their veteran patients. Due to high demand and increasing participation, the program expanded to five sessions per week. Recruitment was largely facilitated by the Tai Chi instructors, who had established exceptionally intimate and trusting relationships with the veteran participants through regular class meetings and were able to verify the inclusion criteria. Furthermore, study announcements were made in each virtual class over several weeks to invite volunteers and provide more information.

Following informed consent, semi-structured interviews were conducted by three of the virtual Tai Chi class instructors. An interview guide with 16 questions was created using questions and probes designed to explore the veterans’ experiences in the Tai Chi program and their perceived benefits. Each interview lasted roughly an hour. All information from the interviews was recorded and transcribed into word documents to ease data analysis and integration with qualitative analysis software. The participants discussed a wide array of detailed experiences with the program, including physical and psychological changes, and the impact on their overall well-being. Data saturation—the theoretical point at which no new emergent themes were observed [40]—was reached after 13 interviews with older veterans who participated in the online inclusive W/A TCC training class in 2023. For additional demographic information about the sample, please see Table 1 and Table 2.

### 2.2. Data Analysis

Building upon standard practices in qualitative data analysis, the data analysis began with “line by line” initial coding. In other words, each line of data was given a unique code. As is common practice in qualitative research, data collection and analysis occurred simultaneously in that the data was collected and then immediately analyzed [41]. As data analysis and data collection progressed, initial codes were gradually refined to form more precise categories or concepts, and through further refinement, emergent themes were produced [42]. To protect against forcing the data to fit into a pre-determined coding structure, a code list was created and remained flexible as new data entered. To ensure inter-coder reliability, initial emerging themes were verified by a second researcher on the project. Both coders have doctorates in social science research, are trained in qualitative methods, and have years of experience, while publishing dozens of articles and books using this method. Throughout the process, the coders evaluated the validity of emerging themes and concepts and reliability within the data. Throughout data collection and analysis, the researchers rigorously discussed and analyzed relevant quotations for themes that addressed the primary research questions. Qualitative data software NVivo 12 (QSR) helped to manage transcripts, to organize categories into themes, and to ensure consistency in coding.

Using constant comparison, codes and emerging themes were analyzed to see if they worked or “fit” in relation to the previously coded extracts and the entirety of the data. Links between themes were explored revealing both the meaning of the themes as well as their relationship with other codes, categories, and other themes [41,42]. Memo writing was also used throughout both the data collection and the analysis process [35]. Memos are written in narrative form and help expand concepts and make connections between various concepts and account for reflexivity, while also allowing the researcher to elevate codes into themes. Lastly, this study has received ethics approval from the Institutional Review Board.

## 3. Results

Our analysis regarding the perceived benefits of adaptive Tai Chi amongst veterans with ambulatory limitations identified four prominent themes. First, participants addressed the perceived psychological benefits of adaptive Tai Chi. Second, participants highlighted the perceived physiological benefits of adaptive Tai Chi. Third, participants discussed the perceived social benefits of adaptive Tai Chi. The findings section below describes how these narratives developed in the qualitative data using excerpts and quotations from the participants. Pseudonyms are used to protect participant confidentiality.

### 3.1. Perceived Psychological Benefits of Adaptive Tai Chi for Veterans

Despite the challenges of adjusting to life as a veteran and coping with disability, the practice of adaptive Tai Chi offered significant cognitive and emotional benefits for our participants. Participants consistently reported a heightened sense of mindfulness, such as maintaining moment-by-moment awareness of thoughts, feelings, bodily sensations, and the surrounding environment with openness, curiosity, and non-judgment. Mindfulness practices, often including breathing methods, guided imagery, and other relaxation techniques, help reduce stress [43]. Like related concepts in Eastern religions like Buddhism, mindfulness fosters awareness of one’s lived experience. Although it was not the exact same process, participants expressed similar outcomes when practicing Tai Chi. For instance, Ella highlights how practicing adaptive Tai Chi helps them focus on the “here and now”, providing psychological benefits.

(I) try to slow down. I actually at night when I’m lying in bed, do the breathing to slow my body down, I imagine doing the motions with my hands. I’ll be lying there still, but I try to link my imagination to my breathing. So that’s probably the only time I’m slow.(Ella)

Ella continues in another passage below to describe the clarity they feel when engaging with Tai Chi.

I guess I’d go with mental, not necessarily mental health, but mental acuity. Just because I have to remember this and I have to make that flow and I make a conscious effort to slow it down and sync the breathing with the hand motions.(Ella)

Ella’s points are corroborated by Cassandra’s perceptions as the mindful component of Tai Chi allows her to slow down when she normally describes her mind as very busy:

I really like when you mentioned moving through water. I think it helps me to really remember to slow down and imagine that there’s some resistance, because otherwise I’m just very busy and can move very fast easily. It reminds me to really slow down and try to flow like water.(Cassandra)

Another participant, Charlie, expressed a similar sentiment regarding mental acuity, mindfulness, and awareness. They also address the modifying effect of the practice via a perceived physiological and mental transformation brought about by the practice.

I think all these practices release all these wonderful chemicals that allow us to heal… It has completely changed my mind and my body until this day.(Charlie)

Mindfulness and awareness are usually combined with narratives of empowerment and a personal sense of control. Individuals express that they have better control over their thoughts, feelings, and emotions because of mindfulness and the Tai Chi practice. Hence, as an additional perceived psychological benefit, participants discussed how the practice of Tai Chi granted them a sense of self-control, which they otherwise struggled to maintain. For example, Dinah below discusses how the practice of Tai Chi has allowed her to have a sense of control over the thoughts and perceptions in their mind.

So now I’m allowing, I’m allowing myself to relax, and when you say drop your shoulders, my shoulders actually drop. So yeah, just to allow myself to pay attention to individual parts of my body. Not just because it’s in pain, but because it’s relaxed and to appreciate what that feels like.(Dinah)

As such, Dinah above addresses the ways in which they can control their attention and focus on things that they wish to focus upon, which in turn brings a sense of relaxation and relief from pain. Another participant, Wayne, who identifies as suffering from post-traumatic stress disorder (PTSD), discusses the ways in which adaptive Tai Chi can help to maintain a sense of emotional regulation, or better perceived control of one’s emotions. Not only does the participant address benefits in terms of PTSD, but they also mention a similar sentiment as above, that is, one of being able to adequately control one’s thoughts and emotions.

(adaptive Tai Chi) Has helped the PTSD quite a bit. Help me calm down a lot. I don’t have near the road rage that I used to have. You know, sometimes I do a mental cloudy hands. So, you know, when somebody’s irritating me I’ll imagine going through that whole thing right, and I don’t know if I’m supposed to do that or not but that’s what I do. So, I’m better.(Wayne)

John below addresses a similar sense of emotional regulation and control and describes it as a mental cleanse.

It’s like a mental cleanse for me. It’s kind of resets everything emotionally.(John)

Lastly, the adaptive Tai Chi program encourages participants to reframe their disability. Our participants were aware of the stigma associated with having a disability within the larger society. For instance, Chet considers the impact of stigma on how they view ability.

I’d rather be the normal guy, rather than the exception. You know, you look down on the two guys that have wheelchairs and one’s got a Walker.(Chet)

As noted above from Chet, “you look down” implies that these messages about the value of ability versus disability are conveyed in subtle and indirect ways. However, adaptive Tai Chi allowed them an opportunity to be active and engage in physical exercise, which was usually unattainable to them. For example, the passage from Dinah below exemplifies this social process.

I think it’s just the shift. Just not giving in to the disability does not my body because I didn’t realize that you could change your brain and how it functions and what it thinks. I didn’t know that that was a thing until taking Tai Chi because a lot of the teachers that you know, I’ve heard [name omitted] say was all about. What is it that you’re telling yourself? You know what are you allowing yourself to believe and just reframing that thinking and just and then that reframing of my thoughts, it’s just really expanded. The ability of what I’ve allowed my body to do. So you know, if it’s a bad day. I give myself, allow myself grace and I didn’t give that. I didn’t know what that was, so now myself, grace.(Dinah)

Thus, the quotation above powerfully illustrates the psychological aspect of living with a disability and how adaptive Tai Chi can help individuals reframe their mindset about perceived limitations. From our participants’ perspective, the practice of Tai Chi offers significant cognitive and emotional benefits for veteran participants, particularly in cultivating mindfulness and present moment awareness. This helped reduce stress and provide clarity by focusing on the “here and now” during the fluid movements. Tai Chi also fostered a sense of mental acuity by requiring focused remembrance of the purposeful motions. As Charlie noted, Tai Chi initiated a “release of wonderful chemicals” that facilitated healing, suggesting they viewed the benefits through a Western medical model of addressing chemical imbalances. The mindful awareness cultivated a deeper sense of self-control, as participants could purposefully relax, manage thoughts, and regulate emotions like PTSD and anger that they previously struggled with. Crucially, participating in adaptive Tai Chi allowed them to view their disability differently and reframe it while also gaining confidence through achievable successes.

### 3.2. Perceived Physiological Benefits of Adaptive Tai Chi for Veterans

Not surprisingly, as a population, veterans and those with disabilities tend to report higher levels of pain and physical discomfort compared to others in the general public [44]. Nonetheless, numerous participants in the study reported an improved sense of physical health and physiological well-being. For one, the practice of Tai Chi reduced their sense of physical limitations and changed the way they describe their disability. In addition, participants discussed changes in their perception of pain, their sense of balance, and improved activities of daily living. Below, Ella explains how Tai Chi has improved her sense of balance and flexibility. Ella also suggests that they engage in balance-based exercises throughout the day and note that their pain has been better managed or has altogether disappeared.

Two out of three pains are gone. Just gone. Isn’t that amazing? It’s interesting. Just last week, my sister and I go to the two yoga classes in the evenings, Monday, Wednesday, and there’s new people and they’re growing and they’re doing this now, Lynn. And I’ve been doing. This for a couple of years. And I remember, Oh my. God, I couldn’t bend that way, so I have. More flexibility and and. I attribute Tai Chi with that also. It’s not just doing yoga. The Tai Chi, I think that Tai Chi has probably increased my balance more than the yoga.(Ella)

Veterans—many of whom faced balance issues due to various conditions—highlighted how adaptive Tai Chi helped them regain stability. Furthermore, they also expressed a sense of progress in their balance and coordination, attributing this improvement to the regular practice of common Tai Chi movements. For instance, Miles addresses a similar sentiment regarding the perceived physical benefits of adaptive Tai Chi to Ella and argues that it helped them to improve their balance in particular. They also exclaim that the practice has improved their eyesight. Of course, we cannot establish whether the practice has physically made a difference to the participant, but we can at least identify the fact that they have perceived a difference.

Yeah I mean some of it, my eyes been healing but just some of the balancing things taught in class. I Try to stand on one foot when I’m cooking…cooking breakfast or some of those things. So, it just I think just that balance. You know it helps. It helps and it’s helped me the most just working through this eye thing which threw me off balance you know. So that’s been the biggest thing I’ve noticed.(Miles)

Lastly, veterans reported significant improvements in their mobility because of participating in the adaptive Tai Chi program. These improvements encompassed various aspects of physical activity, including walking, flexibility, and the ability to perform daily tasks more easily. Some participants would address them when discussing how they incorporated walking and using a cross trainer into their exercise routine, attributing these activities to the influence of the adaptive Tai Chi program. Others, like Chet below, discuss how they started with seated exercises while progressing to standing exercises, demonstrating increased mobility and flexibility over time.

I was particularly interested in that I could do a seated, the stationary seated, and then would be able to progress to a stationary standing.(Chet)

In sum, the participants reported experiencing significant physiological benefits from practicing adaptive Tai Chi, including improved physical health, mobility, balance, flexibility, and pain management. For many veterans with disabilities, the adaptive Tai Chi program helped reduce their perceived physical limitations and change how they viewed their disabilities. The participants described experiencing less chronic pain, with some reporting certain long-standing pains disappearing entirely. They felt that adaptive Tai Chi markedly improved their balance, stability, and coordination, and their flexibility also increased considerably, allowing greater range of motion and ease with activities of daily living. The gentle, low-impact movements helped participants regain strength and dexterity without added injury risks.

### 3.3. Perceived Social Benefits of Adaptive Tai Chi for Veterans

A common concern amongst members of the non-able-bodied community is a sense of isolation due to various structural barriers, which limit social interaction and participation with social institutions in the larger society [45,46]. As such, many individuals face struggles leaving the home or entering social spaces not adequately designed for those with disabilities [47]. Furthermore, the COVID-19 pandemic placed an added burden on individuals in terms of isolation as most individuals were quarantined to avoid spread of the disease [48,49]. Unfortunately, this fact may contribute to a sense of being excluded from society.

Despite the perceived barriers associated with their disability, many participants described how the practice of adaptive Tai Chi increased a sense of social connection to others. As such, they developed a shared identity with many of the teachers and practitioners of the classes. This meant that adaptive Tai Chi was able to provide a sense of belonging that so many individuals crave. Additionally, the shift towards virtual space, not only due to the pandemic, but also due to technological advances in communication technology, means that more individuals within contemporary societies require this stimulation. In general, the participants said that Tai Chi helped to build a strong sense of community, allowed them to share experiences with each other, and provided encouragement and motivation.

In terms of constructing a sense of community, many participants build their common identity as veterans to improve their sense of connection and solidarity with each other. So, in some ways, there is already a sense of identity within the community for this study. However, adaptive Tai Chi helps to strengthen these social bonds. For instance, Charlie highlights the ways in which Tai Chi, and particularly the connection to veterans, strengthens their sense of community.

We have other veterans there. So there’s that connection and also what they bring because we have a ton of expertise.(Charlie)

Two other participants, Sarah and Chet, addressed a similar component of the power of social connections and the interplay with having a veteran status. As such, the use of Tai Chi helps to build strong communal relationships and bonds.

You meet new people and I have adult conversations.(Sarah)

Furthermore, another participant addresses a similar sentiment:

We (referring to veterans) are a family, it is confirmation that they value me, that somebody values me enough to help me adjust. And do the moves that I can do without being able to do before.(Chet)

The participants consistently discussed the ways in which encouragement and motivation from peers played a crucial role in maintaining participation and enthusiasm in the program. Veterans often motivate each other to continue with the classes and achieve their personal goals. In many cases, participation in the program is garnered through word of mouth from friends and colleagues. As in the passages below, the participants address how a colleague helped to encourage them to get involved with the adaptive Tai Chi program.

Interviewer: How did you get involved with this program?

Through another veteran [name omitted for confidentiality]. He was the one who told me what you were doing and how you were doing it so he enticed me to try it out.(Dinah)

In addition, the program had a positive impact on couples who completed the program together. For instance, the quotation below from Miles demonstrates how the adaptive Tai Chi program fosters a sense of community and connection, not only among veterans but also between couples, suggesting that the shared practice can strengthen social bonds outside the practice itself.

Well, it’s just something we can do together. It’s something we have a connection to with that others don’t or whatever, you know. And she’s real studious, I guess about, you know, remembering all the names and moves and everything like that, and it’s just been a good activity for us.(Miles)

As noted above, demonstrations and shared learning experiences within the group help establish a collaborative and supportive environment, one in which veterans feel encouraged to participate and improve. For example, Fiona below mentions that she values the collective experience and the lack of judgment from others.

I appreciate the opportunity. And you guys know, taking the time to help us veterans, you know. Learn tai chi, and have a space where we could do it all together with no judgment.(Fiona)

Furthermore, personal recommendations and encouragement from other veterans are powerful motivators for joining and staying in the program. Thus, these interactions highlight how veterans encourage each other to join and stay committed to the program, fostering a supportive and motivating environment. This peer support system not only helps in individual growth but also strengthens the sense of community within the group.

Lastly, inclusivity is a core principle of the adaptive Tai Chi program, ensuring that veterans feel comfortable and supported regardless of their physical abilities. As such, participants mentioned that the adaptive Tai Chi program emphasizes inclusivity, ensuring that all veterans, regardless of their physical condition, can participate and benefit from the practice. This inclusive approach fosters a welcoming environment for everyone. Charlie below notes that other classes they took required participants to stand, making the environment unwelcoming for those with physical challenges.

When I used to take any other class anywhere that I tried to take a class. Everyone was standing and because I was so physically challenged, I would have to sit…The beautiful thing is if I need to sit because it is adaptive. I can sit.(Charlie)

The program’s structure supports rehabilitation, allowing participants to start with less strenuous seated exercises and gradually progress to standing forms as their strength and mobility improve. Wes addresses a similar sentiment to Charlie regarding inclusivity and the welcoming nature of the adaptive component of the program.

I was interested in using it to help me rehab. I was particularly interested in that I could do a seated the stationary practice and then would be able to progress to a stationary standing.(Wes)

Thus, the adaptive nature of the program allows participants to choose between seated or standing forms based on their comfort and physical ability, making the exercises accessible to everyone.

In sum, the analysis identifies four main themes regarding the perceived benefits of adaptive Tai Chi for veterans. First, participants reported significant psychological benefits, including enhanced mindfulness, emotional regulation, and a sense of control over their thoughts and emotions. This was particularly impactful for those dealing with PTSD. Second, veterans experienced notable physiological benefits, such as reduced chronic pain, improved balance, flexibility, and a changed perception of their physical limitations. Thirdly, Tai Chi fostered a strong sense of social connection among participants, creating a supportive community where veterans could share experiences and motivate each other. Lastly, the inclusivity of the adaptive Tai Chi program was emphasized, allowing veterans with varying physical abilities to participate and benefit from the practice, thereby promoting a welcoming and rehabilitative environment.

## 4. Discussion

The goal of this paper was to explore the perceived benefits of adaptive Tai Chi from the perspectives of veterans with ambulatory limitations. Our paper aimed to answer the following research questions: (a) What are the perceived psychological benefits of adaptive Tai Chi among aging veterans with ambulatory limitations? (b) What are the perceived physiological benefits of adaptive Tai Chi among aging veterans with ambulatory limitations? (c) What are the perceived social benefits of adaptive Tai Chi among aging veterans with ambulatory limitations?

The qualitative analysis revealed several key benefits of adaptive Tai Chi for veterans with ambulatory limitations. First, participants reported psychological benefits, including increased mindfulness, enhanced emotional regulation, and a greater sense of control over their thoughts and emotions, with notable benefits for those managing symptoms associated with PTSD. Second, participants in the study described physiological benefits, such as reduced pain, improved balance and flexibility, as well as a more positive perception of their physical capabilities. Third, the practice of adaptive Tai Chi fostered a strong sense of social connection amongst participants, which created a supportive community where veterans felt comfortable sharing experiences while offering encouragement to each other.

Many of these findings reflect and corroborate the existing literature in terms of the benefits of participating in Tai Chi [25,26]. Of course, other interventions in the academic studies aimed at this population have demonstrated merits in improving physical activity [17,18,19,20,21,22,23]. However, the use of Tai Chi is an additional intervention that may help to address various gaps or limitations in other interventions. First, the novel aspect of our findings lies in its examination of the Tai Chi program’s adaptive structure, a key feature that facilitated the inclusion of veterans with varying physical abilities, allowing them to experience its positive outcomes. Second, the inclusive nature of the program increased a corresponding level of social connection amongst the participants, which was an additional benefit. The adaptive nature of the program also created a redefinition regarding the role of disability in their lives. Third, several participants noted that the adaptive Tai Chi program gave them the opportunity to reframe their disability in an empowering way while expanding what they thought their body could do. Stated differently, adaptive Tai Chi helped participants focus on what they could do, rather than what they could not do (which is common amongst other forms of physical activity), fostering a sense of accomplishment and personal autonomy.

As the United States continues to age and disability continues to create barriers to physical activity among veterans, the need for interventions that are adaptive, affordable, and inclusive becomes paramount. Although not a panacea, our findings indicate that adaptive Tai Chi could be a valuable therapeutic and rehabilitative addition geared towards this population to promote physical activity for veterans with disabilities and improve their well-being. Thus, we suggest that researchers and medical professionals should consider the role of not only Tai Chi as a therapeutic option, but also the importance of inclusion and adaptability.

Despite the evidence regarding the benefits of Tai Chi from previous research and this study, many medical professionals and veterans with disabilities remain largely unaware of the benefits of adaptive Tai Chi for the veteran population. Thus, we suggest that more research opportunities to integrate adaptive Tai Chi into rehabilitation practices are needed. Furthermore, finding ways to further equip VA healthcare providers with the knowledge and resources to confidently discuss and recommend adaptive Tai Chi to their veteran patients would help to disseminate these benefits. This may empower veterans to make informed decisions about their healthcare and positively improve their well-being. Although workshops and seminars are already being conducted, increased informational sessions at VA facilities, community centers, and veteran support groups to showcase adaptive Tai Chi and allow veterans to experience some of the movements may increase participation.

Lastly, the findings of our study offer insights into how adaptive Tai Chi can play a role in reducing the stigma often associated with physical limitations and disability. Interestingly, even though the interview guide did not explicitly ask about societal stigma, one participant brought up their awareness of it and described their own experiences of stigmatization related to their disability (See Chet mentioned on page 10). Despite the sense of stigmatization, participants consistently highlighted the program’s inclusivity—veterans with ambulatory limitations could engage in physical activity without feelings of exclusion or being defined by their disability. We argue that adaptive Tai Chi may reduce the isolating effects of stigmatized notions of disability and encourage greater participation in physical activity. This issue was not directly addressed by the interview guide, and thus we argue that future research could explore and quantify the effects of stigma on this population and the role of adaptive Tai Chi to potentially mitigate that stigma.

We now offer the limitations of the present study. First, there may be concerns about the generalizability of the study findings because the sampling procedure was not random. Further, the participants who participated in the study may have been more motivated to participate in the qualitative research study in the first place, which may have potentially led to a skew in the data in that those who participated had a favorable opinion of adaptive Tai Chi in the first place. However, it should be noted that the strength of qualitative research lies not in the generalizability of the findings, and that is not the goal here, but rather in the detail and depth of the data created from the perspectives of the participants.

Second, in line with the generalizability concerns, sample sizes vary in qualitative studies as data collection should cease only once data saturation has been reached. However, there is a consensus within the qualitative methodological literature that 10 interviews are sufficient for saturation. Most notably, researchers Guest, Bunce, and Johnson used an experimental research design to demonstrate that the threshold for theoretical saturation can occur after as few as 12 interviews [41]. Considering our study entailed 13 interviews, we believe the sample size allowed us to find patterns and thoroughly answer the research question. Thus, we saw no need to continue collecting data once it was clear we met the theoretical threshold.

Third, a significant aspect of this study was the virtual delivery of the adaptive Tai Chi program. The virtual modality of instruction presented limitations including variations in participants’ technological literacy and access to the internet. Extant literature implies that older adults may experience less technological competency and proficiency compared to younger adults [50]. Drawing from the data, one participant (Cassandra) expressed, “virtual is hard,” which insinuates they had difficulty practicing via the virtual modality. However, this was the lone complaint. Research indicates there are an increasing number of older adults accepting and actively engaging in the learning and use of virtual platforms for health-related matters [51,52], and studies imply that virtual instruction may mitigate or alleviate face-to-face participation obstacles [53]. Moreover, studies suggest positive efficacy outcomes with virtual technology related to cognitive and sensory conditions [53]. We believe that by eliminating barriers such as transportation, geographical distance, and the need for access to a physical facility, the virtual format positively impacted the outcome of participants, providing inclusivity for those who lacked reliable transportation coupled with the convenience of participating in the comforts of the participant’s home. As a result, the virtual component is distinctive to our study participants’ overall efficacy.

To conclude, adaptive Tai Chi offers a range of perceived benefits for aging veterans with ambulatory limitations. The program enhances psychological well-being through increased mindfulness, emotional regulation, reframing disability, and a sense of control, but also yields important physiological improvements, such as reduced pain and enhanced physical function. The inclusivity of adaptive Tai Chi programs fosters a strong sense of community, addressing the social isolation often experienced by this population. These findings suggest that adaptive Tai Chi is a promising intervention for enhancing well-being in aging veterans with mobility challenges, highlighting its potential to empower individuals and improve their quality of life.

## Figures and Tables

**Table 1 ijerph-22-01326-t001:** Description of Tai Chi participant sample (N = 13).

Variable	Frequency
*Race*	
White	9
Black	2
Asian	0
Latino	1
Other	1
*Sex*	
Male	7
Female	6
Non-Binary	0
*Age*	
25–35	1
36–45	0
46–55	1
Over 55	7
Not stated	4
*Length of Tai Chi Participation*	
Less than 1 year	2
1–5 years	7
More than 5 years	1
Not stated	3

**Table 2 ijerph-22-01326-t002:** Individual respondent data (N = 13).

Pseudonym	Age	Gender	Level of Mobility Limitation	Duration of Tai Chi Participation
Ella	68	Female	Balance impairment	6 years
Dinah	58	Female	Migraine pain	2 years
Chet	76	Male	Parkinson’s disease	5 years
Miles	51	Male	Dizziness	Not stated
Marcus	70	Male	Difficulty focusing	2 years
Wayne	57	Male	Balance	68 weeks
Cassandra	31	Female	Shoulder stiffness	60 weeks
Nina	68	Female	Low back pain; neck pain	6 weeks plus
Sarah	34	Female	Breathing, leg and back pain	16 weeks
Fiona	Not stated	Female	Knee pain	Not stated
Charlie	Not stated	Male	Unable to raise arm	1 year
John	Not stated	Male	Flexibility	Not stated
Wes	Not stated	Male	Arthritis/surgery	5 years

## Data Availability

Data is contained within the article.

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
