# Peer review of "Perceived Benefits of an Adaptive Tai Chi Program Among Veterans with Ambulatory Limitations"

_ijerph, 2025, doi:10.3390/ijerph22091326_

Round 1

Reviewer 1 Report

Comments and Suggestions for Authors

The manuscript entitled "Perceived Benefits of an Adaptive Tai Chi Program among Veterans with Ambulatory Limitations." explores a meaningful topic with social and clinical relevance, focusing on benefits of an Adaptive Tai Chi Program.

While the topic is interesting, there are some aspects that should be addressed:

  1. Please revise abstract using a conventional format covering background, objectives, methods, results, and conclusions.
  2. I recommend that you should integrate the literature review into the introduction by starting with the statement of the problem, then gradually moving towards the solution approaches. Here, it should be methods to increase physical activity in individuals who use wheelchairs or have ambulatory limitations, and what problems still exist. Additionally, why Tai Chi should be applied to this population, and what gaps still need to be studied, until you state the objective of the current study.
  3. Please add more information regarding the sample size calculation and sampling technique.
  4. Please state more clearly the inclusion and exclusion criteria in the manuscript.
  5. I highly recommend to add more specific details for the methodology of gathering all the information such as the usage of field notes, audio recording, interview instructions.
  6. Please explain and state how to diagnose or screening of impairments or ambulatory restrictions and PTSD.
  7. I highly recommend to demonstrate demographic data of the participants. Including a participant characteristics table including age, gender, level of mobility limitation and duration of Tai Chi participation.
  8. I highly suggest to compare your results with more recent studies, including the rehabilitation interventions for those with ambulatory limitations.

Author Response

Reviewer 1

The manuscript entitled "Perceived Benefits of an Adaptive Tai Chi Program among Veterans with Ambulatory Limitations." explores a meaningful topic with social and clinical relevance, focusing on benefits of an Adaptive Tai Chi Program.

While the topic is interesting, there are some aspects that should be addressed:

  1. Please revise abstract using a conventional format covering background, objectives, methods, results, and conclusions.
    • We’ve revised the abstract to conform to the conventional format. Here is the revised abstract. Background: The growing population of aging veterans in the United States often experiences disabilities that restrict physical activity and limit overall well-being and self-reported health. Accessible, practical, and inclusive interventions are crucial to enhance their well-being. Objectives: This study aimed to explore the perceived benefits of an adaptive Tai Chi program among veterans with ambulatory limitations. Methods: The researchers conducted a qualitative thematic analysis to thoroughly investigate veterans' experiences and perceptions regarding adaptive Tai Chi intervention. Results: Four primary benefits of adaptive Tai Chi emerged from the perspectives of the veteran participants. These included psychological improvements such as heightened mindfulness, enhanced emotional regulation, and a greater sense of control over thoughts and emotions, proving especially valuable for those managing PTSD. Additionally, the program fostered strong social connections and was perceived as highly inclusive, accommodating diverse physical abilities. The accommodating and adaptive nature of the program empowered veterans to reframe their disability and expand their perception of their physical capabilities. Conclusions: These detailed qualitative findings suggest that adaptive Tai Chi may be a valuable therapeutic intervention for improving the overall well-being of aging veterans with ambulatory challenges while also addressing their psychological, social, and physiological needs.
  2. I recommend that you should integrate the literature review into the introduction by starting with the statement of the problem, then gradually moving towards the solution approaches. Here, it should be methods to increase physical activity in individuals who use wheelchairs or have ambulatory limitations, and what problems still exist. Additionally, why Tai Chi should be applied to this population, and what gaps still need to be studied, until you state the objective of the current study.
    • This was perhaps the most helpful comment from all the reviewers, and we decided to make substantial changes to the manuscript as a result. Some of this information was already provided in the literature review so we decided to move this information to the introduction instead. As the reviewer noted, we began with the statement of the problem, highlighting the barriers to physical activity in this population. Next, we include studies regarding various interventions for improving physical activity amongst this population. Then, we discuss the need for interventions that are available for those with ambulatory limitations as well as studies that prioritize reframing disability and self (lines 63-77). Lastly, we included the information regarding Tai Chi and how it may be an additional intervention (lines 79-90).
  3. Please add more information regarding the sample size calculation and sampling technique.
    • As this paper relies upon a qualitative methodology, the sampling methods did not require the use of probability sampling techniques or statistical power calculations. Instead, we relied upon purposive sampling, which we believe improved the quality of the research and the ability to answer the research questions. In short, purposive sampling is a non-probability sampling technique where researchers intentionally select participants based on specific characteristics, knowledge, experiences, or other criteria that are directly relevant to their research question and objectives. In other words, the goal is to acquire “information rich” participants who can provide deep insights into the social phenomena being studied. So, the goal is not statistical generalizability to a larger population. We added more information regarding our sampling techniques to the materials and methods section under 2.1. Please see the highlighted text on page 4 lines 133-172.  
  4. Please state more clearly the inclusion and exclusion criteria in the manuscript.
    • The reviewer is correct that the manuscript requires more information about the inclusion and exclusion criteria. To address the inclusion criteria concern, we added more information about the sampling techniques and a justification for their use. Please see page 5 lines 140-144.
  5. I highly recommend to add more specific details for the methodology of gathering all the information such as the usage of field notes, audio recording, interview instructions.
    • We appreciate the reviewer’s comments. Although field notes can be a useful tool in qualitative methods, we decided they were not required considering the research question is designed to assess the participants’ perspectives, whereas field notes would largely document the social setting of the classes. Interviews included a carefully crafted interview created by the researchers including 16 questions and various opportunities for probes to gather additional information. Please see pages 5 lines 141-170.
  6. Please explain and state how to diagnose or screening of impairments or ambulatory restrictions and PTSD.
    • This project was funded by the Adaptive Sports program of the US Department of Veterans Affairs. All participants were referred to this adaptive Tai Chi program by their healthcare providers at their respective VA medical centers. According to communication with their healthcare providers, all participants have a disability, including emotional distress. We did not utilize any screening instruments within this study to ascertain whether a participant met the psychiatric criteria for PTSD or had received a diagnosis in the past. It is important to note that being diagnosed with PTSD was not one of the inclusion criterion via our methodology. The goal of the study was not to demonstrate the existence or amelioration of psychiatric symptoms or diagnosed conditions, but rather to understand the lived experiences of our participants and their perspectives regarding Tai Chi classes. We gleaned information from the interviews themselves as many of the questions in the interview guide asked about previous impairments or physical limitations, which often included PTSD. In all cases, this information became available during the interview process. Furthermore, the instructors developed a trusting relationship with the participants through several virtual classes and conversations with the participants. Thus, the instructors were able to discern the extent of the ambulatory limitations and any self-reported level of PTSD symptoms or a past diagnosis.
  7. I highly recommend to demonstrate demographic data of the participants. Including a participant characteristics table including age, gender, level of mobility limitation and duration of Tai Chi participation.
  • Thank you for this comment. We provided information about the sample via 2 demographics tables. Please see tables 1 and 2 near the end of the paper on pages 12 and 13. Table 1 includes a description of the sample as a whole and Table 2 includes information about individual respondent data using pseudonyms. As the reviewer suggests, we included information about age, gender, level of mobility limitation, and duration of Tai Chi participation.
  1. I highly suggest to compare your results with more recent studies, including the rehabilitation interventions for those with ambulatory limitations.

We’ve added meta-analyses and reviews describing literature on other interventions for this population. We included this information in the revised introduction (lines 62-76) and the highlighted text in the discussion section, lines 489-495.

Reviewer 2 Report

Comments and Suggestions for Authors

-------------------General Comments

The study adheres to scientific and technical standards. Although the text presents some issues with fluency, these can be easily addressed. However, the introduction may be misleading given the qualitative nature of the study. By the end of the introduction, it creates the impression that the various aspects mentioned would be analyzed quantitatively, which is not the case. Therefore, I see no alternative to demonstrate the relevance of this study other than through the comparison of quantitative data.

-------------------Specific Comments

—Introduction

  • Lines 61–66 – Please provide citations to support this statement.
  • Lines 73–75 – Instead of this unnecessary passage, please present the study’s hypotheses.

-------------------Methods

  • Avoid using the active voice and explanatory language in this section. Only the methodological procedures should be described here.
  • A quantitative analysis of the data is strongly recommended. The qualitative approach alone does not allow the study’s objectives to be fully achieved.

-------------------Results

  • Quantitative data are necessary to meet several objectives of this study. For instance, tools such as NVivo or LIWC may be appropriate in this context.

-------------------Discussion

  • Lines 488–490 – Unfortunately, the qualitative analyses applied do not support this objective. To substantiate these benefits, robust comparative analyses are required.

Author Response

Reviewer 2

The study adheres to scientific and technical standards. Although the text presents some issues with fluency, these can be easily addressed. However, the introduction may be misleading given the qualitative nature of the study. By the end of the introduction, it creates the impression that the various aspects mentioned would be analyzed quantitatively, which is not the case. Therefore, I see no alternative to demonstrate the relevance of this study other than through the comparison of quantitative data.

 -------------------Specific Comments

—Introduction

  • Lines 61–66 – Please provide citations to support this statement.
    •  We have added citations from previous research to support this claim.
  • Lines 73–75 – Instead of this unnecessary passage, please present the study’s hypotheses.
    • Hypotheses aim to assess the relationship between variables. Although necessary in quantitative research, qualitative research uses an inductive method as well as an open-ended flexible approach to explore emerging themes and narratives from non-numerical data. We deliberately chose a qualitative methodology for these reasons and believe this is the strength of the paper as well as an additional way in which it adds to the academic literature. Thus, research questions are pertinent to this section.

 -------------------Methods

  • Avoid using the active voice and explanatory language in this section. Only the methodological procedures should be described here.
    • We have adjusted the language in this section to conform to active voice where necessary. Please see pages 3-5.
  • A quantitative analysis of the data is strongly recommended. The qualitative approach alone does not allow the study’s objectives to be fully achieved.
    • We appreciate the reviewer's insightful comments and their suggestion regarding a quantitative analysis. We understand the value of quantitative approaches in research, particularly for assessing generalizability and statistical relationships. However, the primary goal of this study was to explore the perceived benefits of adaptive Tai Chi from the perspectives of veterans over the age of 65 with ambulatory limitations. As stated in our introduction, our research questions specifically aimed to understand the perceived psychological, physiological, and social benefits. To achieve an in-depth understanding of lived experiences and meaning making, we deliberately chose a qualitative methodology and believe this is the strength of the paper and one way in which it adds to the academic literature. As reviewers 1, 2, and 4 note, it allows us to sufficiently answer the research questions as stated in the introduction.   

-------------------Results

  • Quantitative data are necessary to meet several objectives of this study. For instance, tools such as NVivo or LIWC may be appropriate in this context.
    • The strength of qualitative research lies in providing rich, detailed insights into a social phenomenon from the participants' perspective, rather than via statistical confirmation or generalizable comparative analyses. The methods for this paper aimed to thoroughly explore how veterans perceived these benefits, providing a foundational understanding of their experiences. We reached data saturation after 13 interviews, ensuring that no new emergent themes were observed, which signifies the depth and completeness of our qualitative data for the study's objectives. The detailed narratives and emergent themes identified in our qualitative analysis powerfully substantiate the perceived benefits of the adaptive Tai Chi program from their viewpoints. We offer a justification for the use of qualitative research and the large academic literature supporting its relevance. Please see page 3. Other researchers could certainly build upon these qualitative insights by employing quantitative methods to assess the statistical significance of these benefits across a larger population. However, a quantitative approach falls outside the scope and objectives of this qualitative study. We hope our explanation clarifies the rationale and rigor of our chosen methodology in achieving the stated aims of this paper.  

-------------------Discussion

  • Lines 488–490 – Unfortunately, the qualitative analyses applied do not support this objective. To substantiate these benefits, robust comparative analyses are required.
    • Thank you for suggesting a comparative analysis. Our methodology is designed to achieve an in-depth understanding of lived experiences and meaning making from our participants’ perspectives, directly answering our research questions about perceived psychological, physiological, and social benefits. We maintain that this approach fully achieves the study's stated objectives by providing rich, participant-centered insights.

Reviewer 3 Report

Comments and Suggestions for Authors

This manuscript is well-written and offers insights into an important topic. However, several aspects needed improvement. 

1. Please use language that is free of bias. APA recommends against using "elderly" and suggests using "older adults" or "older population".

2. Two areas of literature (challenges to physical activity and benefits of Tai Chi to aging population) are introduced to explain the merit of the study without being connected. The section 1.2. does not provide information about why Tai Chi is an option to overcome the barriers that were explained in section 1.1. I suggest adding a literature review about online Tai Chi courses.

3. Participants are all from online courses; however, this factor is not adequately reflected in the manuscript. More consideration and discussion about the course being virtual are needed. For example, how did this factor affect the outcomes?

3. Please provide more information about the sample and the coders.

4. The fourth theme (inclusivity) is qualitatively different from the rest of the themes & research questions. I suggest introducing this section with an explanation of why it is essential to include this theme in the manuscript. How does it relate to the research questions?

6. In the discussion section, I suggest discussing the fourth theme before any discussion of implications & interventions.

7. The quotes from participants sometimes fail to provide evidence for the arguments the authors are making. For example, Charlie's quote about wonderful chemicals does not specifically address "the practice changed how they respond to various stimuli". Another quote about feeling like a tiger does not have a social component, but it is used to support the statement, "including various poses, mindfulness, and articulation of flow, veterans find common ground in their physical activities and past experiences, which helps them relate to each other and build meaningful connections". 

8. In section 3.1., the wording "overcome" is used, which implies the medical model of thinking about disability. 

Author Response

Reviewer 3

This manuscript is well-written and offers insights into an important topic. However, several aspects needed improvement. 

  1. Please use language that is free of bias. APA recommends against using "elderly" and suggests using "older adults" or "older population".
  • Thank you for catching this. We have removed all instances of the word, “elderly.”
  1. Two areas of literature (challenges to physical activity and benefits of Tai Chi to aging population) are introduced to explain the merit of the study without being connected. The section 1.2. does not provide information about why Tai Chi is an option to overcome the barriers that were explained in section 1.1. I suggest adding a literature review about online Tai Chi courses.
  • We added substantial changes to the introduction to address this concern. Another reviewer suggested that we combine the literature review into the introduction and so the revised manuscript reflected that change. To address reviewer 2’s comment above, we included information about previous interventions for this population, potential limitations of the past research, and research on the benefits of Tai Chi and how it may potentially address many of these gaps. Further, we added information about the benefits and challenges of a virtual modality for Tai Chi instruction. Please see the revised introduction section beginning on page 2.
  1. Participants are all from online courses; however, this factor is not adequately reflected in the manuscript. More consideration and discussion about the course being virtual are needed. For example, how did this factor affect the outcomes?
  • We find reviewer 3’s recommendation, regarding the virtual, technology component valid. We have addressed the virtual aspects of our study in the discussion section and the introduction as mentioned in comment 2. In brief, we discussed the limitations of the virtual modality, including technological literacy and access to the internet. Despite these limitations, we believe that the virtual modality was a novel feature, improved the feasibility and quality of the research design, and provided a benefit to our participants in the study. By eliminating barriers such as transportation, geographical distance, and the need for access to a physical facility, the virtual modality positively impacted participants by providing inclusivity for those who lacked reliable transportation coupled with the convenience of participating in the comforts of the participant’s home. As a result, the virtual component is distinctive to our study participants’ overall efficacy. Please see page 12 lines 547-562.
  1. Please provide more information about the sample and the coders.
  • We provided information about the sample via 2 demographics tables. Please see tables 1 and 2 near the end of the paper on pages 12 and 13. In addition, we added information in the methods section that describes the coding process, the coders themselves, and the use of multiple coders. Please see the highlighted text on page 4 lines 175-188.
  1. The fourth theme (inclusivity) is qualitatively different from the rest of the themes & research questions. I suggest introducing this section with an explanation of why it is essential to include this theme in the manuscript. How does it relate to the research questions?
  • The reviewer is correct that the inclusion of the theme merits additional explanation. The goal of this paper was to explore the perceived benefits of Tai Chi, including perceived social benefits. In the first submission, the name of the theme was inaccurate. Thus, we changed the name of the third theme to “Perceived Social Benefits,” because it remains consistent with the rest of the themes in the paper and is clearly connected to the research goals. The wording of the 3rd theme in the first submission was an oversight. Thus, we see inclusivity as a social benefit of Tai Chi and hence, it falls under that theme. We see this theme as vital to the paper because it adds an additional way in which Tai Chi can improve the experiences of those in our population.
  1. In the discussion section, I suggest discussing the fourth theme before any discussion of implications & interventions.
  • This comment is slightly related to comment 5, and I hope that the explanation above also addresses the concern in comment 6. In short, we decided to include additional description of the 3rd theme in the discussion section on page 10 lines 485-490. We find this is an additional benefit of the Adaptive Tai Chi program and adds to the extant literature.
  1. The quotes from participants sometimes fail to provide evidence for the arguments the authors are making. For example, Charlie's quote about wonderful chemicals does not specifically address "the practice changed how they respond to various stimuli". Another quote about feeling like a tiger does not have a social component, but it is used to support the statement, "including various poses, mindfulness, and articulation of flow, veterans find common ground in their physical activities and past experiences, which helps them relate to each other and build meaningful connections". 
  • Regarding the first quote from Charlie on line 236, we decided that the use of the word “stimuli” is indeed an inaccurate description. Instead, we changed the language to this passage: “They also address the modifying effect of the practice via a perceived physiological and mental transformation brought about by the practice.” We believe this passage better represents the data. We want to avoid misrepresenting the data as much as possible and appreciate the reviewer catching this inconsistency.
  • Pertaining to the second quote from Charlie, there was a back and forth with how to code this quote during the later stages of the analysis as the main themes began to emerge. The question was whether this quote was evidence of a social benefit or a psychological one. The fact that Charlie chose the word “we” is meaningful because he frames it as something they are doing as a group, even though he himself is also benefiting. However, we decided the reviewer makes a strong point and have removed that quote from the 3rd We believe the quotes already in the manuscript provide ample evidence of the social benefits from the Tai Chi practice.
  1. In section 3.1., the wording "overcome" is used, which implies the medical model of thinking about disability. 
  • The reviewer is correct that the use of the word, “overcome” is a poor word choice. Thus, we decided to take the reviewer’s suggestion and removed it from the manuscript.

Reviewer 4 Report

Comments and Suggestions for Authors

Please see attached review report.

Author Response

Reviewer 4

Reviewer Report for Manuscript ID: ijerph-3703604

Summary The manuscript explores the perceived benefits of an adaptive Tai Chi program among aging veterans with ambulatory limitations. Utilizing a rigorous qualitative approach, authors highlight psychological, physiological, social, and inclusivity benefits derived from the practice. The study effectively addresses a relevant and important area within public health and veterans’ rehabilitation, contributing valuable insights into adaptive therapeutic interventions.

General Comments The manuscript is well-written, structured logically, and contributes meaningfully to existing knowledge. Its strengths include a clear rationale, appropriate qualitative methodology, and detailed thematic analysis providing rich participant insights. The findings align well with existing literature, underscoring the program’s holistic benefits for aging veterans.

Specific Comments Briefly mention potential limitations associated with online delivery, such as reduced interpersonal interaction or technological barriers.

  • Thank you for your time and specific comments. We believe they substantially improve the manuscript. To address this comment, we included a paragraph in the discussion section describing the benefits of using a virtual format as well as the drawbacks from academic literature. Please see the highlighted text on lines 556-572. We also added a section in the introduction addressing some of the same issues. See the highlighted text on page 4, lines 146-152.

Questions for Authors Did participants mention any challenges or limitations regarding the online format of the Tai Chi program?

  • This is a very good question. Although not a question or probe in our semi-structed interview guide, the pros and cons of the virtual nature of the class emerged in the data organically. One participant insinuated that the virtual component created technological challenges. Others addressed the convenience and time-saving benefits of the virtual format. We have added some of this information to the discussion section to help give context to the use of virtual classes. This information is also located in line 556-572.

Recommendations I recommend acceptance of this manuscript after minor revisions addressing the above comments. Please integrate responses to the above question primarily within the "Discussion" section of the manuscript.

Final Decision Accept after minor revisions.

Round 2

Reviewer 1 Report

Comments and Suggestions for Authors

Thank you for your careful and thoughtful revision. I appreciate the effort you made in addressing the previous comments. The manuscript is now clearer and more concise. I have no further suggestions at this time.